# Associations Between GFAP, Aβ42/40 Ratio, and Perivascular Spaces and Cognitive Domains in Vascular Cognitive Impairment

**DOI:** 10.3390/ijms26083541

**Published:** 2025-04-09

**Authors:** Jia Dong James Wang, Yi Jin Leow, Ashwati Vipin, Gurveen Kaur Sandhu, Nagaendran Kandiah

**Affiliations:** 1Lee Kong Chian School of Medicine, Nanyang Technological University, Singapore 308232, Singapore; jwang082@e.ntu.edu.sg (J.D.J.W.); yijin.leow@ntu.edu.sg (Y.J.L.); ashwati.vipin@ntu.edu.sg (A.V.); gurveen.sandhu@ntu.edu.sg (G.K.S.); 2Duke-NUS Medical School, National University of Singapore, Singapore 169857, Singapore; 3National Healthcare Group, Singapore 308433, Singapore

**Keywords:** perivascular spaces, vascular cognitive impairment, blood-based biomarkers, mild cognitive impairment, cerebrovascular disease, neurodegenerative diseases, cognitive dysfunction

## Abstract

Perivascular spaces (PVS) support metabolic clearance in the brain and are increasingly recognized as key contributors to dementia pathogenesis. Plasma-based biomarkers, such as glial fibrillary acidic protein (GFAP) and the amyloid β42/40 (Aβ42/40) ratio, show promise in dementia diagnosis but remain understudied in vascular cognitive impairment (VCI). VCI, a major global cause of cognitive decline, may be more prevalent in Southeast Asia. Despite its impact, it is underdiagnosed compared to Alzheimer’s, highlighting the need for early, reliable markers. This study aims to examine how these biomarkers relate to PVS burden and domain-specific cognitive outcomes in VCI. VCI was defined as global cognition as assessed by a Montreal Cognitive Assessment Score <26, along with the presence of confluent white matter hyperintensities (deep white matter hyperintensities score >2 or periventricular hyperintensities >3), and >1 lacuna. A total of 108 participants (mean age of 67.3 years, 51.9% female) were included. Multivariate ordinal regression assessed biomarker associations with PVS grade, adjusting for age and diastolic blood pressure. A Aβ42/40 ratio <0.05 and GFAP >54.1 pg/mL were used as biomarker thresholds to subgroup the participants, and the relationship between these thresholds and cognitive performance was analyzed. Elevated GFAP (*p* = 0.0438) and a reduced Aβ42/40 ratio (*p* < 0.01) were correlated with a higher PVS grade. In the subgroup with a low Aβ42/40 ratio, a greater PVS burden was associated with poorer executive function (*p* = 0.045, β = 0.612), while in those with high GFAP levels, it was linked to more pronounced impairments in learning and memory (*p* = 0.006, β = 0.375). A lower Aβ42/40 ratio and higher GFAP levels track greater PVS burden in VCI. PVS severity may be associated with domain-specific cognitive decline, highlighting the potential utility of these biomarkers in refining clinical assessments and monitoring disease progression.

## 1. Introduction

Perivascular spaces (PVS) are fluid-filled compartments surrounding blood vessels that facilitate the glymphatic clearance of neurotoxins, including amyloid beta. Dysfunction in PVS drainage has been implicated in the pathogenesis of dementia [1]. On MRI, PVS are typically periarteriolar and linked to arteriopathies and venular dysfunction.

PVS burden increases with age and vascular risk factors, such as hypertension, both of which contribute to dementia pathogenesis. Enlarged PVS may indicate impaired fluid clearance, protein accumulation, or vascular leakage [1]. Recent studies have also shown an association between greater PVS burden and dementia risk, reinforcing its role in disease progression.

Blood-based biomarkers, such as glial fibrillary acidic protein (GFAP) and the amyloid beta 42/40 (Aβ42/Aβ40) ratio, demonstrate promise for early, non-invasive dementia detection. Elevated GFAP, a marker of astrocytic reactivity, may signal blood–brain barrier (BBB) disruption and increased PVS formation. Meanwhile, the Aβ42/40 ratio adjusts for individual amyloid production variability, helping detect pathogenic amyloid accumulation [2,3].

Vascular cognitive impairment (VCI) is one of the most prevalent causes of cognitive decline worldwide, yet it often goes underdiagnosed when compared to conditions like Alzheimer’s disease. Moreover, its incidence is postulated to be higher in Asian populations [4]. Reliable, cost-effective markers are urgently needed to facilitate earlier detection and better management of VCI [4]. A recent meta-analysis identified strong correlations between PVS and age, diastolic blood pressure, and white matter hyperintensities [5]. However, the relationship between PVS and plasma biomarkers in vascular cognitive impairment (VCI) participants remains unclear. This study aimed to characterize these associations and evaluate biomarker profiles (e.g., a low Aβ42/Aβ40 ratio and high GFAP) as potential tools for stratifying cognitive impairment in VCI participants, enhancing clinical phenotyping and diagnosis for such patients.

## 2. Methods

### 2.1. Study Design and Participant Selection

This study utilized cross-section data from the Biomarkers and Cognition Study, Singapore (BIOCIS). Details of the protocols, including demographic data collection, clinical history, physical examinations, cognitive and neuropsychological assessments, blood biomarker analysis, and neuroimaging procedures, have been elaborated in previously published studies [4,6].

Only participants with confluent WMH (a deep white matter hyperintensities score >2 or periventricular hyperintensities >3), >1 lacunae, and a Montreal Cognitive Assessment Score (MoCA) <26 were considered to have VCI as per the published criteria [7].

### 2.2. Neuropsychological and Demographic Protocol

The cognitive assessments used encompassed the Montreal Cognitive Assessment (MoCA), the Visual Cognitive Assessment Tool (VCAT), the Rey Auditory Verbal Learning Test (RAVLT), the Test of Practical Judgment (TOP-J), the Wechsler Adult Intelligence Scale (WAIS) Digit Span Forward, Digit Span Backward, the WAIS Block Design Test, and the Symbol Digit Modalities Test (SDMT). Sleep quality was self-reported by the patients on the Pittsburgh Sleep Quality Index (PSQI).

The participant demographics (age, education years, and gender) were self-reported. Diastolic blood pressure was measured twice, with the average recorded per protocol. The plasma biomarkers analyzed were glial fibrillary acidic protein (GFAP) and the amyloid beta 42/40 (Aβ42/40) ratio. APOE genotyping was performed per protocol.

MRI features were scored by two trained raters using T1 and FLAIR sequences. WMH was graded via the Fazekas scale [6], while the PVS in the basal ganglia and central semiovale were assessed using the Potter scale [8], as detailed in Appendix B.

### 2.3. Statistical Analysis

Statistical analysis of the results was performed using IBM SPSS Statistics 29.

### 2.4. Multivariate Ordinal Correlation

Multivariate ordinal correlation analysis was performed among blood biomarkers, namely GFAP, the Aβ42/Aβ40 ratio, and the PVS grade. This corrected for age, gender, diastolic blood pressure, sleep (as measured by the PSQI), APOE4 status, and WMH severity (as outlined by the Fazekas scale) [6], which have been found to have strong correlations with PVS in previous studies.

### 2.5. Subgroup Analysis of Affected Cognitive Domains

The VCI participants were stratified into subgroups based on biomarker thresholds: high GFAP (>54.1 pg/mL) [9] and a low Aβ42/Aβ40 ratio (<0.05) [10], which were hypothesized to indicate a worse prognosis.

The PVS grade was correlated with cognitive scores using forward stepwise regression, with the PVS grade as the independent variable and neuropsychological test scores as dependent variables. Age, gender, and education were controlled, with Bonferroni correction applied. Significant beta values (*p* < 0.05) were analyzed to identify the most affected cognitive domains.

## 3. Results

### 3.1. Descriptive Statistics of Study Population

Overall, 108 participants were included in this study. Their mean age was 67.32 (standard deviation of 8.28), and 48.1% were males. The means and standard deviations of the demographics (years of education and diastolic blood pressure), neuroimaging scoring (WMH and PVS grade), cognitive test, and biomarkers (GFAP concentration and Aβ42/Aβ40 ratio) are shown in Table 1.

### 3.2. Correlation Between Fluid Biomarkers and PVS

Higher GFAP (odds ratio (OR) = 0.00111, *p* = 0.0438) and a lower Aβ42/40 ratio (OR = −30.8, *p* < 0.01) were associated with a higher PVS grade.

### 3.3. Subgroup Analysis of Association Between PVS Severity and Cognitive Domains Affected

In the subgroup with a low Aβ42/40 ratio (<0.05), an elevated PVS grade was most significantly associated with executive function impairment, as measured by the TOP-J B test (*p* = 0.045, β = 0.612). In contrast, in the subgroup with a high GFAP level (>54.1 pg/mL), a higher PVS grade showed the strongest association with learning and memory deficits, as assessed by the RAVLT test (*p* = 0.006, β = 0.375) and illustrated in Figure 1. No other cognitive tests reached significance in the stepwise regression analysis.

## 4. Discussion

### 4.1. Aβ42/40 Ratio and GFAP Have Strong Associations with PVS Grade

Our findings demonstrate that higher GFAP and a lower Aβ42/40 ratio each correlated with increased PVS burden, aligning with the existing literature. GFAP, a marker of reactive astrocytes, indicates neuroinflammation, leading to cytokine and ROS release, which disrupts the BBB. This disruption allows for fluid accumulation in PVS, increasing the PVS grade [11]. Additionally, reactive astrocytes alter vascular permeability, promoting protein deposition and further PVS enlargement [11].

Similarly, a lower Aβ42/40 ratio reflects higher neuronal Aβ42 accumulation, the pathogenic amyloid variant linked to plaque formation. Since lower fluid Aβ42 suggests increased neuronal retention, this accumulation contributes to PVS enlargement [12]. The Aβ42/40 ratio also enhances diagnostic accuracy by adjusting for individual amyloid production variability.

This is clinically important for VCI patients, as it highlights potentially modifiable astroglial and amyloid-related pathways contributing to small vessel pathology. By identifying patients with elevated GFAP or reduced Aβ42/40 ratios, clinicians may better stratify vascular risk, monitor disease progression, and tailor interventions targeting neuroinflammation or amyloid clearance to slow cognitive decline.

### 4.2. Executive Function Impairment Had Strongest Association with PVS in the Subgroup with a Low Aβ42/40 Ratio

In the participants with a low Aβ42/40 ratio, PVS severity was most strongly tied to executive impairments, specifically in the TOP-J B test, which assesses judgment in real-life scenarios involving medical, financial, social, and safety decisions. Executive function is crucial for these complex judgments.

Higher PVS grades in this subgroup may be linked to preferential Aβ42 accumulation in the posterior parietal cortex, near the centrum semiovale, potentially disrupting the central executive network [13]. Additionally, Aβ42 accumulation in the basal ganglia, which facilitates executive processing in the frontal lobes, may further contribute to impairment [14]. To our knowledge, this is the first study to establish this link in a VCI cohort, emphasizing the unique vulnerability of executive functions under low Aβ42/40 conditions.

Given the role of executive functions in supporting empathy and other aspects of social cognition as assessed by TOP-J, these findings also suggest that PVS-related executive deficits may compromise socioemotional processing [15]. This has implications for functional decline in interpersonal domains, especially as empathy deficits are increasingly recognized in the early stages of VCI. However, further studies are warranted to establish this link [16].

This finding is clinically important, as it underscores the need for early executive function screening in VCI patients with low Aβ42/40 ratios, who may be at greater risk of both cognitive and socioemotional decline. Targeting PVS burden and associated amyloid pathology in this subgroup could help preserve functional independence, particularly in real-world decision making and interpersonal functioning.

### 4.3. Learning and Memory Impairment Had the Strongest Association with PVS in the Subgroup with High GFAP

Among the participants with elevated GFAP, PVS severity most notably correlated with impaired learning and memory, as measured by the RAVLT, which assesses short-term, long-term, and working memory.

GFAP elevation may intensify neuroinflammation and compromise the BBB, amplifying PVS enlargement. Because learning and memory involve widely distributed neural networks, the diffuse impacts of astrocytic reactivity and fluid dysregulation may explain why these domains appear especially vulnerable [17].

This is clinically significant, as it highlights a potential biomarker-driven pathway linking astrocytic reactivity to memory dysfunction in VCI. Identifying patients with high GFAP may allow for earlier detection of those at risk for memory decline, guiding the use of anti-inflammatory or neuroprotective strategies to mitigate cognitive deterioration and preserve daily functioning.

### 4.4. Advantages and Limitations of This Study

The key strengths of this study include its multi-ethnic Asian sample, which enhances the generalizability of our findings, and its focus on VCI—an underrecognized yet prevalent cause of cognitive decline, particularly in Asian populations. Moreover, our community-based recruitment ensured that the results reflect real-world patient profiles rather than a highly selected hospital-based cohort.

The cross-sectional design limits our ability to infer causality. However, a prospective five-year follow-up is underway to validate these findings. The relatively small sample size may also constrain the extrapolation of these results to other settings, highlighting the need for larger confirmatory studies. Moreover, this limits the casual relationship drawn between PVS and various cognitive domains. Furthermore, the absence of recorded data on sleep apnea—a condition strongly associated with PVS burden—represents a potential confounder that may have influenced our findings. In addition, we did not include metabolomics data, which are important to understand the potential mechanism underlying this clinical phenomenon. However, this is likely less clinically relevant as they are not commonly utilized in clinical settings.

### 4.5. Future Directions

Prospective studies should investigate these biomarker–PVS relationships in larger, well-characterized cohorts, such as the UK Biobank or the ADNI. Expanding the biomarker panel to include phosphorylated tau 181 (pTau181) and other emerging markers may further elucidate underlying pathophysiological mechanisms. Additionally, regional PVS analyses (e.g., midbrain vs. basal ganglia vs. centrum semiovale) could reveal domain-specific vulnerabilities. Incorporating advanced neuroimaging tools (e.g., Freesurfer) to quantify PVS volume may provide a more precise correlation with cognitive outcomes. Furthermore, future studies should understand the impact of PVS on empathy, a key domain influenced in VCI.

## 5. Conclusions

Our results highlight that a low Aβ42/40 ratio and high GFAP levels track with greater PVS burden in VCI. Specifically, participants with high PVS burden and a low Aβ42/40 ratio were associated with executive dysfunction, while those with elevated GFAP showed significant learning and memory impairment associations. These findings emphasize the potential of GFAP and the Aβ42/40 ratio as clinically relevant biomarkers for phenotyping VCI, foreshadowing more targeted approaches for risk stratification, monitoring, and early intervention in vascular-related cognitive decline.

## Figures and Tables

**Figure 1 ijms-26-03541-f001:**
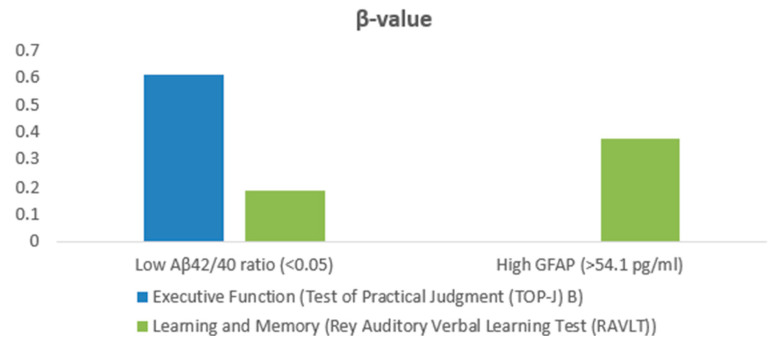
Graph outlining cognitive domains associated with PVS grade in subgroups with a low Aβ42/40 ratio and high GFAP. Abbreviations: GFAP = glial fibrillary acidic protein, Aβ42/Aβ40 = amyloid beta 42/40.

**Table 1 ijms-26-03541-t001:** Demographic, clinical, neuroimaging, biomarker, and cognitive characteristics of VCI participants (N = 108).

Domain	Variable	Mean ± SD
Demographics	Age (Years)	67.32 ± 8.28
	Gender (Male), n (%)	52 (48.1%)
	Years of Education	13.33 ± 4.02
Clinical Measures	Diastolic Blood Pressure (mmHg)	79.67 ± 10.38
Neuroimaging	Fazekas Score (Total WMH Severity)	7.06 ± 2.86
	PVS Grade (Potter Scale)	2.69 ± 1.01
Plasma Biomarkers	GFAP (pg/mL)	94.15 ± 46.60
	Aβ42/40 Ratio	0.058 ± 0.013
Global Cognition	Montreal Cognitive Assessment (MoCA)	23.3 ± 3.31
	Visual Cognitive Assessment Tool (VCAT)	25.3 ± 4.30
Cognitive Subdomains	Rey Auditory Verbal Learning Test (Delayed Recall)	9.29 ± 3.57
	Digit Span Backward (Working Memory)	8.16 ± 2.19
	Test of Practical Judgment—B (Executive Function)	15.2 ± 5.37
	Digit Span Forward (Attention)	10.3 ± 2.17
	WAIS Block Design (Visuospatial)	35.2 ± 8.00
	Symbol Digit Modalities Test (Processing Speed)	60.5 ± 18.3

Abbreviations: GFAP = glial fibrillary acidic protein, Aβ42/40 ratio = amyloid beta 42/40 ratio.

## Data Availability

The original contributions presented in this study are included in the article and Appendix A. Further inquiries can be directed to the corresponding author.

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
