# Peer review of "Associations Between GFAP, Aβ42/40 Ratio, and Perivascular Spaces and Cognitive Domains in Vascular Cognitive Impairment"

_ijms, 2025, doi:10.3390/ijms26083541_

Round 1
Reviewer 1 Report
Comments and Suggestions for Authors
This study’s scope is interesting. However, analysis technique, sample size and data preparation are not enough for acceptance. I hope the authors further study progression.
Table layout is strange.
What is this study’s purpose? If the population of this study participants contain VCI, AD and other types of dementia, we may miss what is evaluated from this study.
There are many factors inducing PVC changes, but the authors did not analyze physiologic and metabolomic factors in the tables. The study’s results may not conclude the specific relationship without analysis of co
The sample size seems too small to say this kind of statistics. There are no distribution graphs of Abeta, GFAP, and PVS grades, and the relationship among them.
The authors evaluated several kinds of cognitive function tests. However, those are not mentioned and shown in the points of relationship with TOP-J and RAVLT. What is the reason why specific tests showed significant correlation?
By your explanation, AD candidate patients may have PVC, then PVC does not necessarily mean the prediction indicator for cognitive decline. Since the study is cross-sectional, it is impossible to say the prediction.
Author Response
Dear Reviewer,
Thank you very much for the opportunity to revise our manuscript. We wish to thank you for taking the time to review our paper and provide comprehensive feedback. We greatly appreciate your insightful comments and suggestions, which will undoubtedly contribute to improving the quality and clarity of our work. Below, we outline the revisions and clarifications made in response to the feedback (in the attached word document) in a point-to-point format.
Thank you!
Regards,
James

Reviewer 2 Report
Comments and Suggestions for Authors
Congratulations. Very relevant and nice study. The scope is very clear and the conduct elegant and sharp
PVS and glinfactic system seem to work on a circadian cycle. Do you think that sleep disturbances could be an adjust variable in the model? Other than hypertension, sleep apnea could diminish the executive function performance in these patients. This is of course an issue of debate but could be discussed in the conclusion.
I should recommend to define the PVS burden more precisely. The same for multi ethnic Asian sample. Does participants have APOE risk polymorphisms tested?
Limitations and future directions are clear and precise.
Is there any relation with apathy one of the most troublesome difficulties in VCI?
Author Response

(The authors gave the same response as above.)

Round 2
Reviewer 1 Report
Comments and Suggestions for Authors
The authors tried to answer the inquiries.
This manuscript is a small size study to see the actual effect in this kind of matter.
This is a cross-sectional study for looking the relationship between PVS and other factors. The authors cannot use the word “predict” and if they want to say the outcome of PVC affecting cognitive function tests and markers. They need to put the point as a limitation in the last part of Discussion.
Author Response
Dear Reviewer,
Thank you very much for the opportunity to revise our manuscript. We wish to thank you for taking the time to review our paper and provide comprehensive feedback. We greatly appreciate your insightful comments and suggestions, which will undoubtedly contribute to improving the quality and clarity of our work. Below, we outline the revisions and clarifications made in response to the feedback in the word document.
